# The Role of Neuropathy Screening Tools in Patients Affected by Fibromyalgia

**DOI:** 10.3390/jcm11061533

**Published:** 2022-03-11

**Authors:** Raffaele Galiero, Teresa Salvatore, Roberta Ferrara, Francesco Masini, Alfredo Caturano, Giovanni Docimo, Margherita Borrelli, Luca Rinaldi, Giovanna Cuomo, Ferdinando Carlo Sasso

**Affiliations:** 1Department of Advanced Medical and Surgical Sciences, University of Campania Luigi Vanvitelli, Piazza Luigi Miraglia 2, I-80138 Naples, Italy; roberta.ferrara@unicampania.it (R.F.); masini.fr@gmail.com (F.M.); alfredo.caturano@unicampania.it (A.C.); giovanni.docimo@unicampania.it (G.D.); margix81@gmail.com (M.B.); luca.rinaldi@unicampania.it (L.R.); ferdinandocarlo.sasso@unicampania.it (F.C.S.); 2Department of Precision Medicine, University of Campania Luigi Vanvitelli, Via De Crecchio 7, I-80138 Naples, Italy; teresa.salvatore@unicampania.it (T.S.); giovanna.cuomo@unicampania.it (G.C.)

**Keywords:** fibromyalgia, small fiber neuropathy, screening, pain

## Abstract

Fibromyalgia syndrome (sFM) is one of the most common causes of chronic pain. This study aimed to assess the presence of small and large fiber impairment in fibromyalgic patients by applying validated scores used in the screening for diabetic neuropathy. The endpoints for the study were the assessment of neuropathy prevalence in sFM patients using the NerveCheck Master (NCM), the Michigan Neuropathy Screening Instrument (MNSI), the Diabetic Neuropathy Symptom (DNS) and the Douleur Neuropathique 4 Questions (DN4). The sample was composed of 46 subjects: subjects with sFM (*n* = 23) and healthy controls (HC) (*n* = 23). The positivity rates in each group for DN4 were significantly different (*p* < 0.001), with a prevalence in symptomatic subjects of 56.3% (*n* = 9) among sFM individuals. A similar difference was also observed with the DNS total score (*p* < 0.001). NCM and MNSI did not disclose significant differences between the two groups. This finding seems to confirm the data regarding the prevalence of a neuropathic pain in sFM patients.

## 1. Introduction

Among systemic diseases and painful extra-articular syndromes, Fibromyalgia or Fibromyalgic syndrome (sFM) is one of the most common causes of chronic pain. This holds true despite various prevalences between countries and ethnic groups, due to diverse diagnostic approaches [1]. sFM is characterized by chronic pain, chronic fatigue, sleep disorders and somatic symptoms, and thus greatly affects the quality of life [1,2]. Patients often receive an accurate diagnosis only several years after the onset of symptoms [1,2]. The global prevalence of sFM has been estimated to be around 2.7%, and is higher in obese females, individuals aged > 50 years and people living in rural areas [2]. In 2016, new criteria were published to better standardize, classify and diagnose the diffused chronic pain of sFM [1,3]. These criteria define pain as generalized if present in four body regions out of the following five: right side, left side, above the waist, below the waist and axial. In addition, the following conditions must be met for diagnosis: pain in at least six of nine possible body sites, sleep disorders or chronic fatigue and symptoms for at least 3 months [1,3,4]. Fatigue is considered both as physical and psychic, and sleep disorders include any or all of the following: difficulty falling asleep, frequent awakenings and restless sleep [1,3,4].

The etiopathogenesis of neuropathy in sFM is still unknown, although many authors claim that “neurobiological stressors” might be involved, especially if present during early life [5]. However, recent biopsy studies have shown evidence of a direct involvement of small unmyelinated C skin fibers and less myelinated Aδ fibers, especially of those linked to thermal-pain sensitivity. Small fiber neuropathy (SFN) is a disease or a damage of the peripheral nervous system, involving sensitive and autonomic small fibers. In one third of cases, diabetes occurs as a recognized cause, and the diagnosis requires at least two altered examinations among biopsy evidence of reduced fiber density, altered perception at quantitative sensory (QST) and altered autonomic testing/clinical questionnaires [6,7,8,9,10]. According to some literature, approximately 50% of patients with sFM have damage to small nerve fibers, thus it is likely that there are other pathophysiological mechanisms still unknown [11,12,13,14]. Moreover, other authors point out that in some cases there is the presence of painful symptoms in absence of any histopathological evidence, thus suggesting an involvement of the central nervous system [15,16,17,18,19]. Based on this data and according to recent literature, our study aimed to assess the presence of a neuropathic impairment in fibromyalgic patients by applying the same validated scores used in the screening for diabetic neuropathy: the Michigan Neuropathy Screening Instrument (MNSI), the Diabetic Neuropathy Symptom (DNS) and the Douleur Neuropathique 4 Questions (DN4) [20,21,22,23,24]. Furthermore, the NerveCheck Master (NCM), a new screening tool with good reproducibility and accuracy in the screening for peripheral neuropathy, has also been employed to specifically evaluate, through cold/warm and heat pain tests, the involvement of small fibers [25,26].

## 2. Materials and Methods

### 2.1. Study Design

This was an observational pilot study conducted on patients recruited at the Internal Medicine ward of I Policlinico, University of Campania Luigi Vanvitelli, from February 2020 to January 2021. Subjects aged < 18 years, pregnant women and people affected by other forms of chronic pain and/or other forms of peripheral neuropathy (e.g., diabetes and specific types of diabetes due to other causes, thyroid dysfunction, Sjogren’s syndrome, sarcoidosis, alcoholic, nutritional and drug-induced/toxic neuropathies) were excluded from the study. The study was approved by our local ethics committee and is in accordance with the 1976 Declaration of Helsinki and its later amendments.

### 2.2. Endpoints

The primary endpoint for the study was the assessment of the prevalence of a neuropathy in the study population. The secondary endpoints aimed to assess: (i) the efficacy of the NCM for detecting a small or a large fiber impairment in sFM; and (ii) the utility of validated scores in diabetic neuropathy (MNSI, DN4 and DNS).

### 2.3. Procedures and Data Collection

Anamnestic data were collected from all patients. Furthermore, all subjects underwent clinical examination, routine biochemical exams (blood count, renal, lipid and liver function, glycemia and glycosylated hemoglobin (HbA1c), C-reactive protein, erythrosedimentation rate and specific rheumatological tests), and following tests and questionnaires: NerveCheck Master, MNSI, DNS and DN4. Moreover, patients with sFM were also presented with a questionnaire on the quality of life (QOL), the Short Form Health Survey (SF-36) and subjected to an evaluation using the Chalder Fatigue Scale and the Pain Visual Analogue Scale (VAS), according to the most recent diagnostic criteria [27,28,29].

#### 2.3.1. Fibromyalgia Assessment

Diagnosis of fibromyalgia was based on anamnestic data and clinical exam, in accordance with most recent guidelines, as discussed in the Introduction [1,3]. To better assess sFM, the SF-36, the Chalder Fatigue Scale and the VAS were administered according to the following methods and evaluations. SF-36 included 36 items, which were differentiated into physical and psychic, and each one evaluated 8 independent scales: functioning, limitations due to physical/emotional problems, vitality (energy/fatigue), emotional well-being, social role, pain, general health status and modification of the same. Then, a profile of each subject was established using the OrthoToolKit and expressed in a percentage from 0 to 100 points (0 = lowest degree of healthy, 100 = optimal) [27]. The Chalder Fatigue Scale consists of 11 items: the first 7 items focus on the exploration of physical fatigue and the remaining 4 focus on psychic fatigue. Based on the results of the questionnaire, fatigue was considered to be absent with a score ≤ 3 and severe if ≥4 [28]. The VAS tool scores perceived pain on a scale from 0 to 10 (0 for absence of pain and 10 for maximum pain), using the following classification: mild (1–3), moderate (4–6) and severe (7–10) perceived pain during the last week/month [29].

#### 2.3.2. Neuropathy Screening

For the evaluation of large and small fiber impairment and to evaluate the presence of symptoms/signs of neuropathy, the following tests were performed. Four different tests were performed using the NerveCheck Master (NCM) tool, evaluating, respectively, vibration, cold/warm and pain sensitivity. A subject was considered to have altered perception with a positive outcome to at least 2 tests. To evaluate vibration perception (VPT), the instrument is placed on the first joint of the big toe, and it emits 9 either true or false stimulations (from 2.7 to 6.4 V). The patient is then asked whether he/she perceives the stimuli or not. The test results are considered abnormal with a score from 0 to 3 and normal within the range 4–12. Regarding cold (CPT), warm (WPT) and pain (PPT) perception, they are evaluated by placing the thermode on the dorsum of the foot. Each of the CPT and WPT tests is repeated 6 times, with fluctuations in temperature (9.8–22.4 °C for cold and 37–44 °C for warm) or empty stimulations. Similarly, the test results are considered abnormal with a score of 0–2 and normal within the range 3–6. PPT assesses the pain threshold through a gradual warm stimulation. At the end of the exam, hyperalgesia is determined if the subject perceives pain before the NCM reaches 35 °C, hypoalgesia is determined if pain is perceived after the NCM reaches 47 °C and values between 36–46 °C determine normal algesia [26].

In addition, the MNSI was performed. It is a validated instrument, generally used when a diabetic peripheral neuropathy (DPN) is suspected. It evaluates the appearance of feet and/or the presence of ulcerations (1 or 0 points by side, whether present or not), the presence of Achilles’ tendon reflex (1 point if absent, 0.5 if reinforced and 0 if present), the perception of vibration stimuli with a 128 Hz diapason and tactile sensitivity to 10 g monofilament (0 if absent, 0.5 if reduced and 1 if present). Peripheral neuropathy was defined as probable with a total score ≥ 2.5 [30].

The Diabetic Neuropathy Symptom Score (DNS) was also employed. It evaluates the presence of sensitive symptoms at the foot level: burning, pain, tingling, numbness and walking instability. For each symptom, 1 point is assigned, if present. The test is positive with the presence of at least 1 symptom [22,23].

Finally, the DN4 questionnaire was used to observe the presence (or lack thereof) of symptoms/signs typical of DPN. Items from 1 to 7 are symptoms (burning, cold, electric shock, tingling, pinprick, numbness, itching), while hypoesthesia to soft touch, to puncture and to skin friction (items 8–10), perceived at foot level, are classified as signs. One point is assigned to the presence of each symptom/sign, and the test is positive with a score ≥ 4 [22,23,24].

### 2.4. Statistical Analysis

The categorical variables were expressed as numbers and percentages, while continuous variables were expressed as median and interquartile range (IQR) or mean and standard deviation (SD), based on their distribution, according to the Shapiro–Wilk test. Between-group differences were assessed either by Fisher’s exact test or the chi-squared test in the case of categorical variables, and by the Student’s t-test or the Mann–Whitney U test in the case of continuous variables, based on their distribution. A *p*-value of 0.05 was considered statistically significant. The relationship of each NCM score to the different tests for fibromyalgia was evaluated by univariate logistic regression models.

Data were analyzed using SPSS version 24 software (SPSS software (IBM, Armonk, NY, USA)) and STATA 15.5 software (StataCorp. 2019. College Station, TX, USA: StataCorp LLC).

## 3. Results

### 3.1. General Characteristics of the Study Population

The study sample consisted of 46 female subjects divided in two groups: subjects with sFM (*n* = 23) and healthy controls (HC) (*n* = 23). Overall, the study population consisted of all female subjects and no statistically significant difference was observed for age. Significantly higher levels of total cholesterol were observed in sFM patients as compared to HC (median 197.5 mg/dL (IQR 178.8–208.8) vs. 170 mg/dL (IQR 153–181.5); *p* = 0.009). LDL cholesterol, though slightly increased in the sFM subgroup, did not significantly change in the two groups. All other parameters were comparable. All data are described in Table 1.

### 3.2. Evaluation of Neuropathy

The study then focused on the evaluation of DPN tests. For the NCM, both the absolute score and the prevalence of positive patients in each test were evaluated. Both assessments did not disclose any significant difference between cases and controls. On the other hand, the positivity rate of each group for DN4 was significantly different (*p* < 0.001), with a prevalence in subjects with altered DN4 of 56.3% (*n* = 9) among sFM individuals. A similar difference was also observed with the DNS total score (*p* < 0.001). Finally, the MNSI did not show significant differences between the two groups, considering both absolute values and prevalence of positivity. All data are reported in Table 2.

### 3.3. sFM and NCM: A Univariate Analysis

The study further assessed the prevalence of positivity for each sFM questionnaire (VAS, Chalder Fatigue Scale and SF-36 questionnaire) in sFM patients. Consistently with the sFM diagnosis, the VAS and the Chalder Fatigue Scale results were elevated, while the SF-36 showed a low total score. All data are reported in Table 3. Then, univariate logistic regression models were employed to assess whether there was a relationship between the findings of each NCM test (in terms of positivity/negativity) and each score used to evaluate fibromyalgia. In most cases, the analysis did not disclose any significant association with the odds ratios set around the unit. All data are reported in Table 4.

## 4. Discussion

This study evaluated, for the first time, the utility of the NCM and other validated diabetic neuropathy screening scores in a population of sFM patients. In this setting, several authors have described the presence of an impairment of myelinated Aδ and small unmyelinated C fibers, thus defining a picture of small fiber pathology (SFP) [14,15,31,32,33].

The NCM is a new tool able to detect signs of both large (by VPT) and small fiber impairment (by CPT/WPT and heat pain test). No significant difference emerged between HC and sFM patients. This result seems consistent with previous studies outlining QST in subjects with fibromyalgia, showing that there is not always a difference between patients with and without SFP, thus suggesting that SFP could have a negligible impact on somatosensory system function [18]. Moreover, several studies on diabetic neuropathy have shown that quantitative sensory/thermal tests are characterized by variable sensitivity and produce discordant results [26,34]. Thus, the instrument could not match the efficacy of more specific methods, such as skin biopsy and nerve conduction studies. Finally, as this is a new screening tool which so far has only been employed for diabetic subjects, the NCM sensibility and specificity could be too low, compared to skin biopsy or other QST, to provide the same results in an sFM population as well.

Originally, the study set out to evaluate whether the MNSI, the DNS and the DN4, generally used to screen other conditions of neuropathy, could be useful to detect symptoms and signs of neuropathy in sFM.

The MNSI allows for a large fiber neuropathy to be screened and is used to diagnose a probable DPN in diabetic subjects [9,30]. Thus, to exclude a large fiber neuropathy in our population, the team decided to adopt this score to screen sFM patients [9,35,36]. In the sample size, there was no difference between the two groups regarding the presence of signs of neuropathy screened by the MNSI. This result seems consistent with the literature, which shows that in sFM patients, a large fiber neuropathy is absent [9,18,22,35]. During this study, however, a nerve conduction study (the gold standard to evaluate the presence of large fiber neuropathy) has not been carried out to confirm this hypothesis [37].

The DN4 and the DNS are validated scores used to detect neuropathic pain and screen a peripheral neuropathy in diabetic patients [23,24,33]. Moreover, the DN4 has previously been employed in an sFM population with a good association with more specific methods, as well as laser-evoked potentials [38,39,40]. In our study, a significant difference between the two groups emerged by evaluating the presence of neuropathic pain at DN4, and screening a possible neuropathic pathology at DNS, with a prevalence of an altered score in almost 50% and in 20% of sFM patients, respectively. The neuropathic pain detected according to DN4 could be related to a possible neuropathy or driven by central mechanisms [15,16,17,18]. The DNS presents a significative difference with HC, as also observed in diabetic subjects in a validation study showing that this questionnaire had good diagnostic accuracy in the screening for diabetic polyneuropathy [38,39,40]. These results suggest that the DN4 and the DNS may be useful to determine a neuropathic pain also in sFM patients [9,39]. However, the absence of a diagnostic confirmation with a proper algorithm or validated tools, as suggested by the most recent guidelines, does not allow us to better verify the origin of the pain detected by DN4 and DNS [8,9,10]. The etiology of neuropathic pain in Fibromyalgia is still unknown and much debated, but many authors suggest a central and peripheral impairment in addition to the manifestation [31]. In some studies, authors evaluated it through more specific investigations, such as skin biopsy laser-evoked potentials, thus confirming the peripheral involvement [17,32]. We can assert that some patients from our sFM population are also affected by painful signs and symptoms, possibly due to a pain of neuropathic origin [13]. Moreover, by evaluating symptoms, the DNS could be a useful score not only to screen patients, but also to distinguish among different categories, based on the expected response to drugs (e.g., SNRI-Duloxetine/Milnacipran and Pregabalin).

All the other tests used to evaluate psychic characteristics (VAS of pain, Chalder Fatigue Scale and QOL questionnaire) were not associated with peripheral neuropathy scores. This observation could support the hypothesis for a different etiopathology of these phenomena [31].

This study presents several limitations. Primarily, the small sample size, which does not allow to generalize the results of the observations. Secondly, there is a lack of a more specific tool to define the etiopathogenesis of the pain detected at DN4 and DNS, such as nerve conduction studies or skin biopsy, and of a comprehensive QST protocol to compare the results of NCM. Finally, the lack of a dedicated questionnaire, such as the small fiber neuropathy—symptoms inventory questionnaire (SFN-SIQ), which would have been useful to compare our results [41].

## 5. Conclusions

The NCM and the MNSI, which were used for the first time in the evaluation of a neuropathy in sFM subjects, do not show significant differences compared to HC. These observations could be the expression in the absence of small and large fiber functional impairment as detected by the NCM, and from no signs of neuropathy at the MNSI in our cohort of sFM subjects. On the contrary, the DN4 and the DNS seem to show the presence of typical neuropathic pain in sFM subjects as well. However, the absence of a neurophysiological assessment and/or of skin biopsies did not allow us to better compare these results and to define the etiopathogenesis of the pain.

Further studies are needed to better define the impairment of fibers in these patients and to clarify the role of scores validated in other conditions.

## Figures and Tables

**Table 1 jcm-11-01533-t001:** General characteristics of the study population (*n* = 46).

	Fibromyalgia (*n* = 23)	Controls (*n* = 23)	*p*
Sex (M/F), *n* (%) Age (yrs.), median [IQR]	0 (0)/23 (100) 45 (33–54)	0 (0)/23 (100) 53 (33–61)	1.000 0.317
BMI (kg/m^2^), median [IQR]	27.6 (23–30.7)	25.3 (20.8–33.4)	0.619
Blood Pressure (mmHg), median [IQR]			
Systolic	100 (95–140)	125 (110–140)	0.380
Diastolic	75 (65–80)	70 (70–80)	0.862
Heart Rate (bpm), median [IQR]	72 (67.5–75.5)	75 (64–86)	0.816
Glycemia (mg/dL), median [IQR]	87 (84–95)	80.5 (73.5–95.3)	0.132
HbA1c (%), median [IQR]	5.8 (5.6–6)	5.7 (5.4–5.8)	0.134
Creatinine (mg/dL), median [IQR]	0.71 (0.66–0.82)	0.70 (0.61–0.80)	0.579
eGFR (mL/min), median [IQR]	96.6 (80.3–108.8)	101 (77.8–112)	0.706
Cholesterol (mg/dL), median [IQR]			
Total	197.5 (178.8–208.8)	170 (153–181.5)	0.009
HDL	55 (42.8–56.6)	50 (40.5–61.5)	0.804
LDL	110.5 (85–125)	91 (83.5–125)	0.481
Triglycerides (mg/dL), median [IQR]	115 (77–225)	110 (84.5–126)	0.592

Abbreviations: BMI: body mass index; eGFR: estimated glomerular filtrate rate; HDL: high density lipoproteins; LDL: low density lipoproteins IQR: interquartile range.

**Table 2 jcm-11-01533-t002:** Diabetic Peripheral Neuropathy (DPN) tests.

	Fibromyalgia (*n* = 23)	Controls (*n* = 23)	*p*
NCM, median [IQR]			
Vibration	2.5 (1–9)	4 (0–9)	0.765
Cold	6 (3–6)	6 (5–6)	0.695
Warm	4.5 (1–6)	4 (1–6)	0.715
Pain	1 (0–1)	1 (1–1)	0.587
NCM, *n* (%)			
Vibration	12 (54.5)	10 (43.5)	0.556
Cold	3 (13.6)	1 (4.3)	0.346
Warm	8 (36.4)	6 (26.1)	0.530
Pain	9 (40.9)	8 (34.8)	0.763
MNSI, mean (SD)	0.57 (0.73)	0.57 (0.68)	0.953
MNSI, *n* (%)			n.a.
Positive	-	-	
Negative	23 (100)	23 (100)	
DN4, mean (SD)	4.13 (2.45)	0.86 (1.04)	<0.001
DN4 Positive, *n* (%)	9 (56.3)	-	<0.001
DNS, mean (SD)	2.57 (1.45)	0.35 (0.71)	<0.001

Abbreviations: NCM: NerveCheck Master; SD: standard deviation; MNSI: Michigan Neuropathy Screening Instrument; DNS: Diabetic Neuropathy Symptom; DN4: Douleur Neuropathique 4 Questions; IQR: interquartile range; n.a.: not applicable.

**Table 3 jcm-11-01533-t003:** Tests for fibromyalgia (*n* = 23).

Fibromyalgia Test	Fibromyalgia (*n* = 23)	Controls (*n* = 23)	*p*
VAS scale, median [IQR]	8 (7.1–8.8)	-	n.a.
Chalder Fatigue Scale, median [IQR]	23 (19.8–25.8)	-	n.a.
SF-36 Questionnaire, median [IQR]			
(1) Physical activity limitations due to health disorders	45 (25–65)	-	n.a.
(2) Social activity limitations due to physical/emotional disorders	25 (0–25)	-	n.a.
(3) Limitations in common role activities due to physical problems	33.3 (0–66.7)	-	n.a.
(4) Body pain	30 (22.5–40)	-	n.a.
(5) General Mental Health (psychological stress and wellness)	48 (36–60)	-	n.a.
(6) Limitations in common role activities due to emotional problems	37.5 (25–50)	-	n.a.
(7) Vitality (strength and fatigue)	22.5 (22–43.8)	-	n.a.
(8) Overall perception of general health status	35 (27.5–47.5)	-	n.a.

Abbreviations: VAS: Visual Analogic Scale; IQR: interquartile range; n.a.: not applicable.

**Table 4 jcm-11-01533-t004:** Relationship between NerveCheck Master and fibromyalgia scores (*n* = 23).

	NCM Vibration	
	OR (95% IC)	*p*
VAS Scale	0.70 (0.23–2.12)	0.530
Chalder Fatigue Scale	1.29 (0.93–1.80)	0.126
SF-36 Questionnaire		
(1) Physical activity limitations due to health disorders	0.97 (0.92–1.02)	0.969
(2) Social activity limitations due to physical/emotional disorders	0.98 (0.95–1.02)	0.984
(3) Limitations in common role activities due to physical problems	0.99 (0.96–1.02)	0.985
(4) Body pain	0.94 (0.85–1.04)	0.206
(5) General Mental Health (psychological stress and wellness)	0.97 (0.88–1.06)	0.435
(6) Limitations in common role activities due to emotional problems	1.01 (0.95–1.07)	0.704
(7) Vitality (strength and fatigue)	1.00 (0.93–1.07)	0.929
(8) Overall perception of general health status	0.95 (0.88–1.02)	0.135
	NCM Cold	
VAS Scale	2.80 (0.30–26.15)	0.367
Chalder Fatigue Scale	0.92 (0.69–1.22)	0.551
SF-36 Questionnaire		
(1) Physical activity limitations due to health disorders	0.96 (0.89–1.03)	0.241
(2) Social activity limitations due to physical/emotional disorders	0.97 (0.90–1.04)	0.393
(3) Limitations in common role activities due to physical problems	1.00 (0.96–1.03)	0.820
(4) Body pain	0.96 (0.86–1.07)	0.463
(5) General Mental Health (psychological stress and wellness)	0.97 (0.86–1.09)	0.582
(6) Limitations in common role activities due to emotional problems	0.98 (0.91–1.06)	0.631
(7) Vitality (strength and fatigue)	0.94 (0.84–1.05)	0.282
(8) Overall perception of general health status	0.89 (0.76–1.04)	0.132
	NCM Warm	
VAS Scale	0.65 (0.21–2.04)	0.462
Chalder Fatigue Scale	0.99 (0.77–1.25)	0.901
SF-36 Questionnaire		
(1) Physical activity limitations due to health disorders	0.92 (0.84–1.01)	0.073
(2) Social activity limitations due to physical/emotional disorders	1.00 (0.96–1.04)	0.908
(3) Limitations in common role activities due to physical problems	0.99 (0.96–1.02)	0.987
(4) Body pain	0.99 (0.91–1.08)	0.894
(5) General Mental Health (psychological stress and wellness)	1.08 (0.98–1.20)	0.126
(6) Limitations in common role activities due to emotional problems	0.95 (0.88–1.02)	0.139
(7) Vitality (strength and fatigue)	0.99 (0.92–1.06)	0.716
(8) Overall perception of general health status	0.98 (0.92–1.05)	0.524
	NCM Pain	
VAS Scale	0.42 (0.09–1.92)	0.265
Chalder Fatigue Scale	0.97 (0.76–1.24)	0.805
SF-36 Questionnaire		
(1) Physical activity limitations due to health disorders	0.98 (0.94–1.03)	0.417
(2) Social activity limitations due to physical/emotional disorders	0.96 (0.91–1.02)	0.187
(3) Limitations in common role activities due to physical problems	1.00 (0.97–1.02)	0.714
(4) Body pain	0.98 (0.90–1.06)	0.577
(5) General Mental Health (psychological stress and wellness)	1.05 (0.96–1.15)	0.292
(6) Limitations in common role activities due to emotional problems	0.98 (0.92–1.04)	0.441
(7) Vitality (strength and fatigue)	0.93 (0.85–1.03)	0.156
(8) Overall perception of general health status	0.95 (0.87–1.03)	0.188

Abbreviations: OR: odd ratio (OR); NCM: NerveCheck Master; VAS: Visual Analogic Scale (VAS).

## Data Availability

Not applicable.

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
