# Peer review of "The Role of Neuropathy Screening Tools in Patients Affected by Fibromyalgia"

_jcm, 2022, doi:10.3390/jcm11061533_

Round 1

Reviewer 1 Report

The paper by Galiero and colleagues aimed at verifying the validity of screening tools approved for Diabetic Peripheral Neuropathy in detecting neuropathy in patients with Fibromyalgia.

I’m afraid the paper presents several weaknesses that prevent the publication.

The authors have dealt with a very sensitive issue too superficially:

Major concernes

  • Whether it is true that roughly the 40% of patients with FMG have abnormal skin biopsies, its functional role is still controversial. For this reason, experts agreed in defining those alterations as small fibre pathology (SFP) rather than small fibre neuropathy (SFN). Please refer to the studies and commentaries by Uceyler (Brain, 2013), Clauw (Pain,2015) Oaklander (Pain,2016) Fasolino (Pain,2020).
  • The definition of small-fibre neuropathy is inaccurate as well as the diagnostic criteria. Please refer to the studies by Devigili (Brain, 2008;2019), Terkelsen (Lancet Neurology 2017)
  • The nerve check master (NCM) seems to be a short (and less precise) version of the QST. The QST is considered a valid diagnostic tool to disclose SFN, particularly the thermal-pain thresholds: how can the author suggest the presence of a small fibre neuropathy with no such abnormalities?
  • The DN4 is a valid screening tool to detect neuropathic pain, not small fibre neuropathy. The authors should have relied on dedicated questionnaires such as the SFN-SIQ.
  • Besides the above mentioned inaccuracies, the normality of clinical-based tools such as MNSI and NCM and the abnormalities of the DN4 and the DNS would have suggested the absence of clinically relevant abnormalities together with the presence of sensory signs, a condition not sufficient to define a neuropathy.

I strongly suggest to carefully review the literature on the topic and rethink the discussion of the paper. What they author found is, basically, the evidence of pain (well known in FMG); the chance for this pain to be neuropathic according to DN4 (already known and not implying a neuropathy, neuropathic pain could be driven by central mechanisms); the absence of clinically relevant signs as suggested by previous authors (Clauw; Fasolino). I would not conclude for the presence of a neuropathy, especially if detected with the DN4!

Author Response

  • Whether it is true that roughly the 40% of patients with FMG have abnormal skin biopsies, its functional role is still controversial. For this reason, experts agreed in defining those alterations as small fibre pathology (SFP) rather than small fibre neuropathy (SFN). Please refer to the studies and commentaries by Uceyler (Brain, 2013), Clauw (Pain,2015) Oaklander (Pain,2016) Fasolino (Pain,2020).

Re: We wish to thank the reviewer for his/her precious comment, which allowed us to improve the paper. Accordingly, we used, where possible, the definition of small fiber pathology (SFP) instead of small fiber neuropathy (SFN). Moreover, we added the requested references.

  • The definition of small-fibre neuropathy is inaccurate as well as the diagnostic criteria. Please refer to the studies by Devigili (Brain, 2008;2019), Terkelsen (Lancet Neurology 2017)

Re: We wish to thank the reviewer for his/her precious comment, which allowed us to improve the paper. Accordingly, we modified the definition and diagnostic criteria of small fiber neuropathy and added the requested references. (Line 55-58)

  • The nerve check master (NCM) seems to be a short (and less precise) version of the QST. The QST is considered a valid diagnostic tool to disclose SFN, particularly the thermal-pain thresholds: how can the author suggest the presence of a small fibre neuropathy with no such abnormalities?

Re: We wish to thank the reviewer for his/her precious comment, which allowed us to improve the paper. Accordingly, we modified the text by explaining this result. (Line 210-219)

  • The DN4 is a valid screening tool to detect neuropathic pain, not small fibre neuropathy. The authors should have relied on dedicated questionnaires such as the SFN-SIQ.

Re: We wish to thank the reviewer for his/her precious comment, which allowed us to improve the paper. Accordingly, we modified the text by adding this point in to Limitations and by explaining why we used DN4 Line (271-273)

  • Besides the above-mentioned inaccuracies, the normality of clinical-based tools such as MNSI and NCM and the abnormalities of the DN4 and the DNS would have suggested the absence of clinically relevant abnormalities together with the presence of sensory signs, a condition not sufficient to define a neuropathy.

Re: We wish to thank the reviewer for his/her precious comment, which allowed us to improve the paper. Accordingly, we modified the definition of small fiber neuropathy in introduction section and modified the text in conclusion section. Line 55-58 and 277-286.

I strongly suggest to carefully review the literature on the topic and rethink the discussion of the paper. What they author found is, basically, the evidence of pain (well known in FMG); the chance for this pain to be neuropathic according to DN4 (already known and not implying a neuropathy, neuropathic pain could be driven by central mechanisms); the absence of clinically relevant signs as suggested by previous authors (Clauw; Fasolino). I would not conclude for the presence of a neuropathy, especially if detected with the DN4!

Re: We wish to thank the reviewer for his/her precious comment, which allowed us to improve the paper. Accordingly, we modified the text especially for what concerns Discussion and Conclusion sections.

Reviewer 2 Report

This study assessed the presence of neuropathy in patients with fibromyalgia by applying validated scores used in the screening of diabetic neuropathy.

In general:

This is an interesting study, as it’s results provide clinicians with user-friendly and validated tools to screen patients with FM for neuropathy, such as the DN4 and the DNS.

The authors recognize that the sample sizes were rather small and that the patients were relatively young.

Nevertheless, the conclusions of the study are relevant.

I have few remarks.

Methods

2.3.1 FM assessment

The VAS tool scores perceived pain on a scale from 0 to 10 (0 for absence of pain and 10 for maximum pain), using the following classification: mild (1-3), moderate (4-6), and severe (7-10) [22].

This was retrospective study. When was this score recorded? Pain assessment using this scale represents only a snapshot and it is generally known that pain levels in fibromyalgia vary significantly from day to day and even during the day.

Wouldn’t it have been better to use a mean score of the maximum, minimum, and usual pain intensity during the last week or month?

Discussion

Maybe, the discussion would be clearer if the text was structured in paragraphs, rather than in a continuous text.

References:

Last reference: I don’t think it’s numbered (33.?)

Author Response

In general:

This is an interesting study, as its results provide clinicians with user-friendly and validated tools to screen patients with FM for neuropathy, such as the DN4 and the DNS.

The authors recognize that the sample sizes were rather small and that the patients were relatively young.

Nevertheless, the conclusions of the study are relevant.

 I have few remarks.

Methods

2.3.1 FM assessment

The VAS tool scores perceived pain on a scale from 0 to 10 (0 for absence of pain and 10 for maximum pain), using the following classification: mild (1-3), moderate (4-6), and severe (7-10) [22].

This was retrospective study. When was this score recorded? Pain assessment using this scale represents only a snapshot and it is generally known that pain levels in fibromyalgia vary significantly from day to day and even during the day.

Wouldn’t it have been better to use a mean score of the maximum, minimum, and usual pain intensity during the last week or month?

Re: We wish to thank the reviewer for his/her precious comment, which allowed us to improve the paper. Accordingly, we modified the text by better clarifying this point in Methods section Line 115

Discussion

Maybe, the discussion would be clearer if the text was structured in paragraphs, rather than in a continuous text.

Re: We wish to thank the reviewer for his/her precious comment, which allowed us to improve the paper. Accordingly, we modified the text in Discussion section

References:

Last reference: I don’t think it’s numbered (33.?)

Re: We wish to thank the reviewer for his/her precious comment, which allowed us to improve the paper. Accordingly, we added the reference number

Round 2

Reviewer 1 Report

The authors partially improved the manuscript as compared with the previous version, but the text still presents important flaws.

The major issue is the diagnosis of SFP: authors did not perform a proper diagnostic algorithm nor used validated tools to diagnose SFP, therefore I suggest to remove SFP from the title and focus the paper on the role of the different tools used in FMG patients. Furthermore, I don’t get why the authors look disappointed by their findings regarding the NCM, while a large part of the literature agrees about the absence of functional impairment in FMG patients. As previously suggested, based on the authors’ finding I would conclude that in their cohort of FMG patients there is no functional impairment of small and large fibres based on the NCM and no signs of neuropathy as disclosed by the MNSI. The DN4 detected neuropathic pain in these patients, however the authors cannot conclude about the origin of this pain. SFP should be removed from the manuscript and only cited in discussion.

Other comments:

How did the authors enrolled the control population? Age and sex-matched?

SFN is present in 49% of the cases studied: which cases? FMG cases?

sFM can be classified in different subgroups based on the objective evidence of painful SFP or dysautonomia, even though these are often present together. I think this sentence can be removed because the authors are not dividing their cohort in different subgroups.

and evaluations for possible/probable SFP, according to the following tests and questionnaires: Nerve Check Master, MNSI, DNS and DN4. How can the authors be able to evaluate the presence of SFP without a validated diagnostic tool (skin biopsy; QST)? As I have reported in my previous revision, the Nerve check master seems to be a shorter version of the QST and according to the methodology described, an alteration of the thermal pain thresholds could be considered diagnostic. Though it should be clarified that the sensibility and sensitivity of this tool have never been tested in comparison with gold standards. Please remove the SFP-evaluation paragraph, there is no information about clinical signs and normality of nerve conduction study, two mandatory conditions for SFP diagnosis.

As a suggestion, if you think your sample size is too small it would be better to improve it rather than citing it as the cause of a negative results, becasue it seriously affects the impact of the paper.

Author Response

The authors partially improved the manuscript as compared with the previous version, but the text still presents important flaws.

The major issue is the diagnosis of SFP: authors did not perform a proper diagnostic algorithm nor used validated tools to diagnose SFP, therefore I suggest removing SFP from the title and focus the paper on the role of the different tools used in FMG patients.

Re: We wish to thank the reviewer for his/her precious comment, which allowed us to improve the paper. Accordingly, we modified the title.

Furthermore, I don’t get why the authors look disappointed by their findings regarding the NCM, while a large part of the literature agrees about the absence of functional impairment in FMG patients. As previously suggested, based on the authors’ finding I would conclude that in their cohort of FMG patients there is no functional impairment of small and large fibres based on the NCM and no signs of neuropathy as disclosed by the MNSI.

Re: We wish to thank the reviewer for his/her precious comment, which allowed us to improve the paper. Accordingly, we modified the Conclusion section. Line (207-209, 223-224 and 268-275).

The DN4 detected neuropathic pain in these patients, however the authors cannot conclude about the origin of this pain.

Re: We wish to thank the reviewer for his/her precious comment, which allowed us to improve the paper. Accordingly, we modified the Conclusion section. Line (238-243 and 272-275).

SFP should be removed from the manuscript and only cited in discussion.

Re: We wish to thank the reviewer for his/her precious comment, which allowed us to improve the paper. Accordingly, we modified the text.

Other comments:

How did the authors enroll the control population? Age and sex-matched?

Re: Yes, the study population results were equally distributed for age and sex. Accordingly, we added in Table 1 a line also for sex results. Line 164-165 and Table 1.

SFN is present in 49% of the cases studied: which cases? FMG cases?

Re: We wish to thank the reviewer for his/her precious comment, which allowed us to improve the paper. Accordingly, we modified the text. Line 60-61.

sFM can be classified in different subgroups based on the objective evidence of painful SFP or dysautonomia, even though these are often present together. I think this sentence can be removed because the authors are not dividing their cohort in different subgroups.

Re: We wish to thank the reviewer for his/her precious comment, which allowed us to improve the paper. Accordingly, we removed the sentence.

and evaluations for possible/probable SFP, according to the following tests and questionnaires: Nerve Check Master, MNSI, DNS and DN4. How can the authors be able to evaluate the presence of SFP without a validated diagnostic tool (skin biopsy; QST)? As I have reported in my previous revision, the Nerve check master seems to be a shorter version of the QST and according to the methodology described, an alteration of the thermal pain thresholds could be considered diagnostic.

Re: We wish to thank the reviewer for his/her precious comment, which allowed us to improve the paper. Accordingly, we modified the text. Line 91-92.

Though it should be clarified that the sensibility and sensitivity of this tool have never been tested in comparison with gold standards. Please remove the SFP-evaluation paragraph, there is no information about clinical signs and normality of nerve conduction study, two mandatory conditions for SFP diagnosis.

Re: We wish to thank the reviewer for his/her precious comment, which allowed us to improve the paper. Accordingly, we modified the text. Line 112-114 and 174.

As a suggestion, if you think your sample size is too small it would be better to improve it rather than citing it as the cause of a negative results, because it seriously affects the impact of the paper.

Re: We wish to thank the reviewer for his/her precious comment, which allowed us to improve the paper. Accordingly, we removed from the paper sentences citing small sample size as a cause of negative results.

Round 3

Reviewer 1 Report

The authors improved the manuscript to the previous version, however there are still some minor issues that deserve their attention:

Abstract:

“This finding seems to confirm the data regarding the prevalence of a neuropathic pain in sFM patients suggesting that similar mechanisms to those of nerve impairment in diabetic patients, could be also present in sFM subjects.” The statement about a similarity of mechanisms between FMG and diabetic patients is not supported by the results. I would remove the second part of the sentence suggesting that similar mechanisms to those of nerve impairment in diabetic patients, could be also present in sFM subjects.

Discussion

“The neuropathic pain detected according to DN4 could be related to a possible neuropathy, also driven by central mechanisms”. It is not clear what do the authors mean with neuropathy driven by central mechanism. I guess they were referring to neuropathic pain, I would write “or” instead of “also”.

thus suggesting the hypothesis of similarities between diabetic and sFM SFP”. This sentence is not supported by the results obtained, the authors did not perform any validated test for SFP. I would remove this sentence.

This result could be due to the absence of any association between central and peripheral damage. This sentence is vague an confusing, I suggest to remove it.

and of other QST” . A comprehensive QST protocol?

“the lack of dedicated questionnaire small fiber neuropathy” The lack of a dedicated questionnaire such as the small fibre neuropathy etc..

“These observations could be the expression of the absence of a small and large fiber functional impairment at the NCM”  the expression of the absence of a small and large fibre functional impairment as detected by  the NCM

“On the contrary, the DN4 and the DNS, which was used for the first time in the evaluation of sFM” It is not the first time that the DN4 has been used in FMG patients. Anyway it should be were and not was.

Author Response

The authors improved the manuscript to the previous version, however there are still some minor issues that deserve their attention:

Abstract:

This finding seems to confirm the data regarding the prevalence of a neuropathic pain in sFM patients suggesting that similar mechanisms to those of nerve impairment in diabetic patients, could be also present in sFM subjects.” The statement about a similarity of mechanisms between FMG and diabetic patients is not supported by the results. I would remove the second part of the sentence suggesting that similar mechanisms to those of nerve impairment in diabetic patients, could be also present in sFM subjects.

Re: We wish to thank the reviewer for his/her precious comment, which allowed us to improve the paper. Accordingly, we removed the second part of the sentence. Line 29-31.

Discussion

“The neuropathic pain detected according to DN4 could be related to a possible neuropathy, also driven by central mechanisms”. It is not clear what do the authors mean with neuropathy driven by central mechanism. I guess they were referring to neuropathic pain, I would write “or” instead of “also”.

Re: We wish to thank the reviewer for his/her precious comment, which allowed us to improve the paper. Accordingly, we modified the text. Line 239.

thus suggesting the hypothesis of similarities between diabetic and sFM SFP”. This sentence is not supported by the results obtained, the authors did not perform any validated test for SFP. I would remove this sentence.

Re: We wish to thank the reviewer for his/her precious comment, which allowed us to improve the paper. Accordingly, we removed the sentence. Line 244.

This result could be due to the absence of any association between central and peripheral damage. This sentence is vague an confusing, I suggest to remove it.

Re: We wish to thank the reviewer for his/her precious comment, which allowed us to improve the paper. Accordingly, we modified the text. Line 258, 259.

and of other QST”. A comprehensive QST protocol?

Re: We wish to thank the reviewer for his/her precious comment, which allowed us to improve the paper. Accordingly, we modified the text. Line 263.

“the lack of dedicated questionnaire small fiber neuropathy” The lack of a dedicated questionnaire such as the small fibre neuropathy etc..

Re: We wish to thank the reviewer for his/her precious comment, which allowed us to improve the paper. Accordingly, we modified the text. Line 264, 265.

“These observations could be the expression of the absence of a small and large fiber functional impairment at the NCM” the expression of the absence of a small and large fibre functional impairment as detected by the NCM

Re: We wish to thank the reviewer for his/her precious comment, which allowed us to improve the paper. Accordingly, we modified the text. Line 272.

“On the contrary, the DN4 and the DNS, which was used for the first time in the evaluation of sFM” It is not the first time that the DN4 has been used in FMG patients. Anyway it should be were and not was.

Re: We wish to thank the reviewer for his/her precious comment, which allowed us to improve the paper. Accordingly, we modified the text. Line 273, 274.